

# Research on automatic matching of online mathematics courses and design of teaching activities based on multiobjective optimization algorithm

Jiafeng Li[1], Lixia Cao[1] and Guoliang Zhang[2]

[1] Rizhao Polytechnic, Rizhao, China
[2] Rizhao Donggang District Education and Sports Bureau, Rizhao, China

## ABSTRACT

The teaching of the optimization algorithm is a new kind of swarm intelligence optimization technique, which is superior in optimizing many simple functions. Still, it is not evident in processing some complex problems (group and teaching classification). Achieving automatic matching and knowledge transfer in online courses is imperative in mathematics education. This study proposes a design scheme MTCBO-LR (multiobjective capability optimizer-logistic regression), based on multitask optimization, which enables precise knowledge transfer and data interaction among many educators. It incorporates the standard TLBO algorithm to optimize, provides a variety of learning tactics for students at different stages of mathematics instruction, and is capable of adaptively adjusting these strategies in response to actual teaching needs. Experimental results on various datasets reveal that the proposed method enhances searchability and group diversity in various optimization extremes and outperforms similar methods in resolving to multitask teaching problems.

# INTRODUCTION

The current trend in educational tasks is increasingly focused on personalized and specific development, aiming to meet the unique educational needs of each student and achieve dynamic resource allocation (*Ong & Gupta, 2016*). Therefore, the school education system should continue to explore and improve teaching methods. Intelligent teaching-assistant systems are a solution that can automatically allocate appropriate learning resources and tasks based on students' learning situations and performances, achieving dynamic resource allocation. Meanwhile, teachers should pay attention to and analyze students' personalized needs, stimulating their learning interests and potential through diversified teaching methods and strategies. This personalized and specific teaching approach not only helps improve students' learning effectiveness but also cultivates their innovation and problem-solving abilities, better adapting to the needs of future society (*Huang, 2021*).

From design modelling to task optimization, viewing mathematics education as a multiobjective optimization issue that can be solved by iteratively refining a set of objectives under constrained decision-making variables is gaining traction as a promising

Corresponding author
Jiafeng Li, rzljf@outlook.com

new approach. Evolutionary multitask, or multi-factor optimization, is a new emerging sub-field in optimization that combines the ideas of evolutionary computation and multitask learning. Unlike the previous two, multi-factor optimization integrates multiple interrelated objective functions to solve various optimization tasks simultaneously and explore potential relationships among tasks to improve efficiency and effectiveness (*Yu & Zhao, 2019*). Mathematics teaching is a four-satellite activity. Students can perform multiple tasks. To effectively utilize the commonalities and differences among various problems, knowledge of one job should be used to assist the solution of another job, which is a multiobjective problem-solving process.

Existing algorithms typically solve singular optimization tasks in isolation and seldom leverage knowledge gained from one task to help solve another (*Yu, Lan & Jiang, 2022*). However, it is plausible that correlations exist between diverse problems, potentially augmenting the efficiency and effectiveness of problem-solving. Drawing inspiration from the human ability to manage multiple tasks, *Gupta, Ong & Feng (2015)* employed multitask learning in evolutionary computation, introducing a novel optimization problem category known as multi-factor optimization and a multi-factor evolutionary algorithm. The objective of this algorithm is to exploit potential synergies among disparate optimization problems, making efficient use of both shared and distinct characteristics of the various issues. This article introduces the MTCBO framework, which is founded on multitask optimization. The MTCBO algorithm is based on the principles of teaching optimization algorithms. It capitalizes on the fact that students can transfer knowledge from teachers or classmates with similar or dissimilar attributes, effectively leveraging the commonalities between problems. This approach has the potential to greatly enhance the efficiency and performance of problem-solving.

We propose MTCBO-LR, which can realize the function of introducing different learning strategies at different stages. Three learning strategies are presented to update knowledge at the teaching level, and two are introduced to update knowledge at the student level. The main contributions of this article are:

(1) Using multiple strategies can maintain the diversity of the population while maintaining a higher convergence accuracy and speed. The algorithm adaptively selects appropriate learning strategies, enhances the positive influence of shared information, and improves the overall optimization ability of the algorithm.

(2) The normal distribution perturbation strategy is introduced to help students escape from local optimality. The selection strategy based on the ranking is adopted to find high-quality solutions while maintaining the population's diversity, ensuring the algorithm's generalization and accuracy.

## RELATED WORK

The complexity of mathematics teaching tasks is continuously increasing, leading to increased complexity in teaching preparation and student acceptance of the teaching process. To address these challenges, researchers have proposed the concept of multiobjective optimization. By leveraging multiobjective optimization techniques, it is possible to transfer knowledge across diverse tasks, thereby improving overall problem-

solving capabilities. Researchers have made significant progress in implementing multiobjective optimization algorithms and have devised effective methods to leverage information among tasks.

For instance, *Bali et al. (2020)* presented a novel evolutionary computing framework that can learn online and enhance optimization. *Zhou, Feng & Tan (2021)* proposed the MFEA-AKT algorithm, which utilizes adaptive knowledge transfer to match appropriate crossover operators adaptively. *Gupta, Ong & Feng (2015)* introduced the multi-factor evolutionary algorithm, which combines multitask optimization with evolutionary algorithms. *Liang et al. (2019)* proposed a novel multi-factor evolutionary algorithm with a genetic transformation strategy that can improve knowledge transfer efficiency.

These approaches facilitate exploring and exploiting subspaces for each task and the combined search space, enabling individuals to discover lesser-known areas and enhance optimization performance. The advent and evolution of multiobjective optimization algorithms offer effective techniques and concepts for tackling mathematical teaching tasks.

Research into multitask evolutionary algorithms has received considerable attention to lessen the computational load; *Huang (2021)* proposed the SAEF-AKT framework, which uses an adaptive knowledge transfer strategy and creates a proxy model by mining past searches for information. Similarly, *Wu, Zhu & Chen (2022)* introduced the MFEA/D-DRA algorithm that employs decomposition and dynamic resource allocation strategies to transform the problem into several sub-problems for adaptive resource allocation. Furthermore, *Bali et al. (2020)* presented the cognitive evolutionary multitask engine, which analyzes the data produced during multitask optimization and lowers optimization costs by adjusting the degree of online genetic transfer.

Some investigations have proposed novel algorithms and frameworks to expedite the optimization process. For instance, the collaboration protocol evolutionary framework introduced by *Chen, Chen & Peng (2021)* can partition the problem into low-dimensional subproblems and use a locally-search algorithm grounded on quasi-Newton methods to achieve knowledge exchange and local search for solving high-dimensional optimization problems. Similarly, *Hao, Qu & Liu (2020)* developed the EMHH algorithm, a graph-based evolutionary multitask hyperheuristic algorithm that addresses multitask problems through cooperative action. Additionally, *Xu & Zhang (2019)* presented the MTO-FWA algorithm, which employs transfer sparks to transmit genetic information to enhance optimization efficiency. Furthermore, Feng et al. integrated multi-factor and differential evolution algorithms to resolve multiobjective optimization problems. *Rao, Savsani & Balic (2012)* devised a heuristic swarm intelligence optimization algorithm named Teaching-Learning-Based Optimization (TLBO), which optimizes by emulating classroom teaching processes. TLBO outperforms traditional swarm intelligence optimization algorithms with fewer control parameters, a more straightforward overall structure (*Chen, Chen & Peng, 2021*; *Abdel-Basset, Mohamed & Chakrabortty, 2021*), faster-running speed, and more straightforward implementation. Consequently, it has been widely adopted in various fields, capturing the attention of many researchers and continually improving (*Li, Gong & Wang, 2020*; *Shukla, Singh & Vardhan, 2020*). The MTCBO-LR algorithm

incorporates the normal distribution perturbation strategy. After generating new descendant students, the MTCBO-LR algorithm utilizes a rank-based selection strategy to retain students with superior quality.

## METHOD

### MTCBO

The EMO algorithm plays a significant role in scientific research and engineering applications (*Wang & Pei, 2019*). Therefore, solving multiobjective optimization problems is of great importance. Here, we first define EMO:

$$\min F(x) = (f_1(x), f_2(x), \dots, f_m(x)) \tag{1}$$

The EMO algorithm is a method for solving optimization problems with multiple conflicting objective functions by seeking numerous optimal solutions to balance the relationship between these objective functions. EMO algorithms are widely used in scientific research and engineering applications, such as machine learning, logistics planning, power system optimization, *etc*. They can help people better understand and solve practical problems.

The present study introduces an enhanced single-objective multitask optimization algorithm, MTCBO. The primary iteration of this algorithm consists of two scenarios. During the initial stages of the algorithm, if the number of solutions in the archive set Arc is less than Na, *i.e.*, Arc is small in size, it is challenging to estimate the trend of the Pareto Front (PF) accurately. We propose the MTCBO approach for optimizing the MOP method (*Nia, Ghaffari & Zolnouri, 2022*) to address this. This technique uses crossover and mutation operators on P populations to create new offspring populations. The proposed method subsequently updates the archive set Arc and applies fast, non-dominated sorting and crowded distance selection to choose the next generation population in the population.

These two individuals are selected from different sub-populations if this probability condition is unmet. Finally, we update the archive set Arc and sub-populations p1, p2,…,

$$\{x_1, x_2, \dots, x_k\} = \{\arg \min P_1(x_1), \arg \min P_2(x_2), \dots, \arg \min P_K(x_K)\} \tag{2}$$

### MTCBO-LR

#### *Adaptive knowledge transfer*

To improve the MTCBO algorithm, we propose two methods of adaptive knowledge transfer for improvement. Each student chooses learning strategies from the two strategies according to their situation, and students update knowledge through appropriate learning strategies in different optimization stages. Different from the greedy selection strategy of the original TLBO algorithm, the selection of new offspring is based on individual factor value and factor diversity.

Following the above principles, we divided all students into different classes, each class was assigned different learning strategies based on skill factors, and the performance of the strategies varied. Among them, teacher $X_{teacher,\tau}^f$ and $X_{teacher,n}^f$ were the best individuals

from class and class factor value, respectively. Teaching assistants $X_{k,\tau}^m$ and $X_{k,n}^m$ were randomly selected individuals from class $P_\tau$ and class $P_n$, respectively. Mean and were the average scores of class $P_\tau$ and class $P_n$, respectively. The individual update formula is as follows:

$$X_{i\_new,\tau\_new} = \begin{cases} X_{i,\tau} + rand(1,D) \cdot * \left( \left( X_{teacher,\tau}^f + X_{k,\tau}^n \right)/2 - TF * X_{mean,\tau} \right) & \text{if } rand >= rmp \\ X_{i,\tau} + rand(1,D) \cdot * \left( \left( X_{teacher,n}^f + X_{k,n}^m \right)/2 - TF * X_{mem,n} \right) & \text{otherwise} \end{cases} \quad (3)$$

X Strategy 1: This strategy aims to increase diversity among each student. Teachers and teaching assistants provide instruction based on the difference between their average level and the class's average score.

$$X_{i\_nww,\tau\_new} = \begin{cases} X_{i,\tau} + rand(1,D) \cdot \left( \left( X_{teacher,\tau}^f + X_{k,\tau}^n \right)/2 - TF * X_{meam,\tau} \right) & \text{if } rand >= rmp \\ X_{i,\tau} + rand(1,D) \cdot * \left( \left( X_{teacher,n}^f + X_{k,n}^m \right)/2 - TF^* X_{mean,n} \right) & \text{otherwise} \end{cases} \quad (4)$$

X Strategy 2: A reverse learning mechanism is implemented in this learning strategy.

$$\ddot{X}_i = U + L - X_i \quad (5)$$

The value of RMP allows for balancing between exploration and exploitation of the search space. When RMP (*Badi, Mahapatra & Raj, 2023*) is close to 1, students will always receive knowledge transfer from individuals with the same attribute, which can help scan critical areas of the search space but increase the risk of falling into local optima. Conversely, when rmp is lower than 1, communication between individuals from different cultures can aid in escaping local optima. Figure 1 illustrates the algorithm flow of MTCBO-LR, which follows the above principles.

## Adaptive matching and vertical spread strategies

In the later stage of learning, students are likely to encounter learning bottlenecks that make it difficult to continue improving. Through a certain degree of "perturbation", students can be helped to break out of the bottleneck, explore a wider search space, and find better solutions. The criterion that initiates this strategy is as follows: when the optimal value of any task factor remains unchanged for ten successive generations, all students will undergo perturbation, and their knowledge will be updated according to their respective learning strategies. The formula for updating individuals is as follows:

$$X_{i\_new,\tau\_new} = X_{i,\tau} + \text{normrnd}\,(0, 0.01, 1, D) \quad (6)$$

In the later stages of learning, students are likely to encounter learning bottlenecks that are difficult to overcome. By introducing a certain degree of "disturbance", students can break through bottlenecks, explore a broader search space, and find better solutions (*Zhao, Di & Cao, 2021*). The trigger condition for this strategy is: when the optimal factor value of
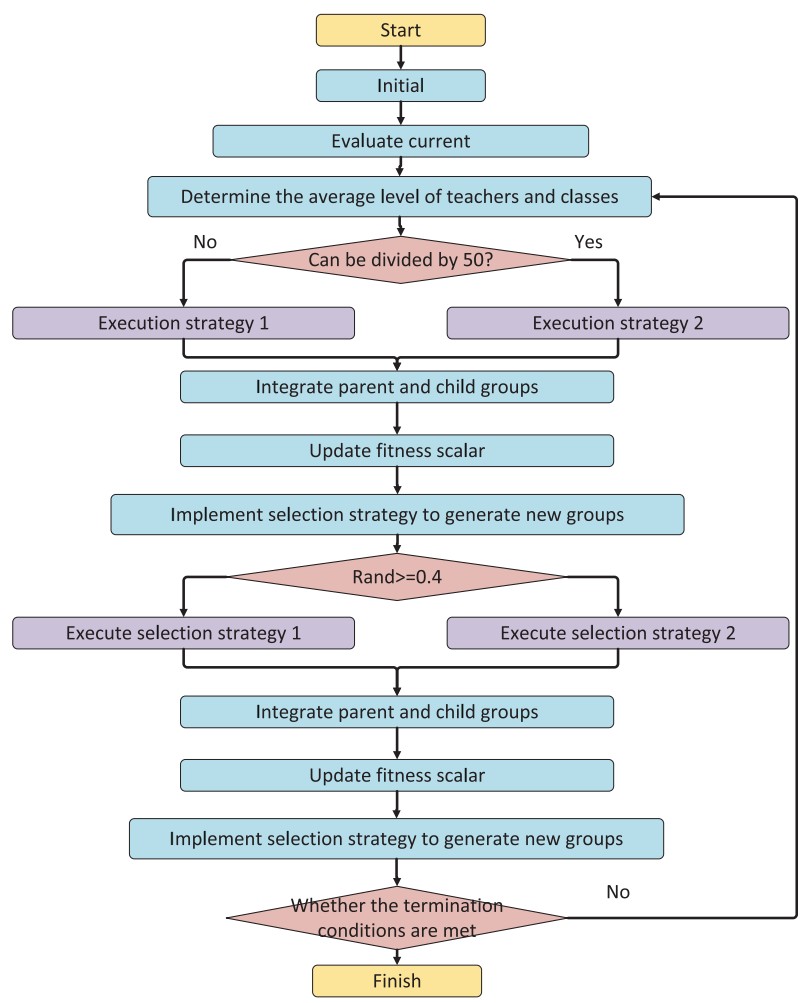

**Figure 1  The algorithm flow of MTCBO-LR.**     

any task does not change for ten consecutive according to the learning strategy. The individual update formula is as follows:

Where ormrnd (0, 0.01, 1, D) is a $1 \times D$ normal distribution matrix, 0 and 0.01 are the mean and standard deviation, respectively.

The MTCBO-LR method refers to two MFEA algorithms, the original MFEA and MFEA-AKT. Unlike the original MFEA, which uses a single crossover operator, MTCBO-LR uses multiple crossover operators with different search performances for knowledge transfer. Specifically, during the algorithm execution process, MTCBO-LR randomly selects two parent individuals to generate offspring individuals. If these two parent individuals have the same skill factor, the SBX crossover operator is used for crossover operation. Otherwise, whether to perform crossover or mutation operation on these two parent individuals is determined according to the preset random mating probability. Suppose parents have different skill factors and need to perform crossover operations. In that case, MTCB0-LR will randomly assign cf for activating the crossover factor W or p and use the corresponding crossover operator to perform crossover operations on W and

p. In addition, the generated individuals, *i.e.*, migration offspring, will use the crossover factor Cfa as their attribute. If two offspring C1 and C2, are generated by a crossover operation or a mutation operation without knowledge transfer, then C1 and C2 will inherit their corresponding parent crossover factor, respectively.

$$c_1 = \left( p_1^1, p_1^2, \ldots, p_2^i, p_2^{i+1}, \ldots, p_2^j, p_1^{j+1}, \ldots, p_1^n \right)$$
$$c_2 = \left( p_2^1, p_2^2, \ldots, p_1^i, p_1^{i+1}, \ldots, p_1^j, p_2^{j+1}, \ldots, p_2^n \right)$$

(7)

## Selection strategy based on KNN

The KNN model classifier was trained using a historical transfer data set containing positive and negative transfer individuals (*Rahman et al., 2022*), denoted as HTS. Let M represent the set of all trained KNN model classifiers. The following steps were taken to train the KNN model classifier: (1) A similarity matrix was constructed based on the distance between individuals in HTS. The initial label of all training data was set to "unlabeled." The maximum local neighbour N of each individual in HTS, which covers the maximum number of neighbours of the same category, was calculated. (2) The maximum local neighbour Ni of all individuals Si with the "unlabeled" label was put into set Q. (3) The maximum value N in set Q was found, and a model M = <Cls(s;), Sim(s;), Num(s;), Rep(s;)>, where Num(s;) = N, was constructed. This indicates that s; covers the maximum number of neighbours of the same category. Model M was then put into the model set M.

Using the KNN classifier based on the above steps can avoid the problem of randomly selecting individuals from the current population, which often leads to repeat visits to hopeless regions in the search space and low efficiency in traditional algorithms. Mathematical proofs suggest that introducing a conflicting solution and its opposite has greater potential than introducing two unrelated random solutions to approach the optimal global solution without prior information (*Tiwari & Chaturvedi, 2022*). Therefore, we defined a conflicting point in this article. Additionally, a selection strategy was designed that considers both factor value and factor diversity to update individuals.

In this article, we design a selection strategy considering factor value and diversity to renew individuals. It is implemented in the MTCBO-LR algorithm. For the populations corresponding to task T, we arranged them from most minor to most significant regarding factor value and assigned Fitness Rank (FR) to each individual. The calculation formula of FR is as follows:

$$FR_j^i = i, \quad i = 1, 2, \cdots, NP$$

(8)

The smaller the factor value of an individual, the lower its corresponding fitness level. The population corresponding to task Tyj is sorted in ascending order of factor diversity, and the resulting order is the diversity rank (DR) of the individual, which is defined by the formula:

$$DR_j^i = i, \quad i = 1, 2, \cdots, NP \tag{9}$$

The smaller the factor diversity of an individual, the smaller the corresponding diversity level. In task Tyj, the rankRy of individual Xyi is defined as:

$$R_j^i = \omega \cdot DR_j^i + (1 - \omega) \cdot FR_j^i \tag{10}$$

where $\omega = \dfrac{G}{Maxgen}$ is the current iteration number and Maxgen is the maximum iteration number.

# THE EXPERIMENT

## Experiment settings

This experiment mainly verifies the optimization effect of the proposed method on small-scale problems, which is suitable for forming small-class online collaborative learning groups. A simulated data set containing seven features of 30 learners (numbered S1–S30) was selected.

To assess optimization test problems, this study selected seven sets of classic single-objective to multitask optimization problems and ten sets of complex single-objective multitask optimization problems for analysis. Each benchmark test problem consisted of two component tasks containing a single-objective optimization task. The seven classic multitask optimization technique reports had varying degrees of overlap and similarity between the component tasks. The overlap degree refers to the similarity or difference of the optimal global solution of the two tasks in the same search space, with three types of overlap degrees. CI meant that the component tasks in the benchmark test problem had precisely the same global optimal solution. PI meant that the optimal global solution of the component tasks in the benchmark test problem was partially the same. NI meant that the component tasks in the benchmark test problem had completely different global optimal solutions. The similarity degree refers to the similarity of the shape and size of the two tasks in the same search space. It should be noted that since the MTCBO-LR algorithm processed two tasks once in each iteration, while the SOTLBO algorithm processed only one task, the total number of iterations of the SOTLBO algorithm was 250 when reaching the termination condition, while the MTCBO-LR algorithm was 500.

## Single-objective task benchmarking problem performance

Assuming the mean and standard deviation were 0 and 0.01, respectively, the other parameters were the same as the original publication. Table 1 presents the average factor values and standard deviations of five algorithms independently run 20 times, with the best experimental results highlighted in bold. The final comparison was based on the average results of 20 independent runs.

As shown in Table 1, the MTCBO-LR algorithm performs better in seven tasks. Specifically, the MTCBO-LR algorithm has a minor score in all group problems in the seven tasks and has a clear advantage, especially compared to the traditional SOTLBO algorithm. This is because the MTCBO-LR algorithm employs multiple strategies to

**Table 1 Score performance of seven groups of single-target multi-task benchmarks.**

| MCI+HS | MFEA-GHS | EMT | TMOMFEA | MTCBO-LR |
|---|---|---|---|---|
| MCI+MS | −0.063 | −1.512 | −0.523 | −0.621 |
| MCI+LS | 2.965 | 1.412 | −1.523 | −1.421 |
| MPL+HS | 1.508 | 0.052 | 1.762 | −2.402 |
| MPI+MS | −0.107 | −0.145 | −1.422 | −0.091 |
| MPI+LS | 2.991 | 3.379 | 2.764 | −0.987 |
| MNO+HS | 0.215 | −0.445 | 0.124 | −0.447 |

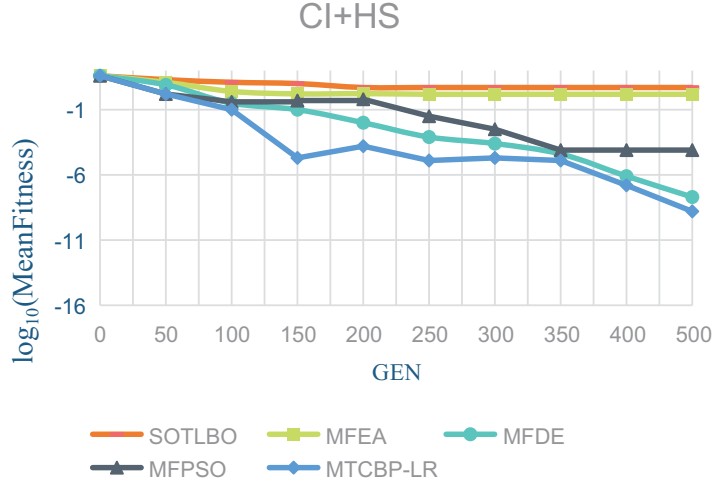

**Figure 2 The effectiveness of five algorithms on CI+HS tasks.**

maintain population diversity, thus maintaining high convergence accuracy. The superiority of the MTCBO-LR algorithm demonstrates that novel and adaptive learning strategies can reduce the negative transfer of knowledge and that rank-based selection strategies can maintain population diversity while finding high-quality solutions, enhancing the algorithm's development and exploration capabilities. The experimental results are confirmed in Figs. 2–4, indicating that the MTCBO-LR algorithm has high effectiveness in solving MTO problems.

## Multiobjective task benchmark performance

To assess the efficacy of the MTCBO-LR algorithm on multiobjective tasks, we conducted a comparative analysis across five distinct task combinations. To ensure parity across all trials, we maintained a population size of 100 and a maximum evaluation threshold of 200,000 as the termination criterion (*Liu & Wang, 2019*). Each algorithm underwent 20 independent runs for the sake of comparison. This study employed the iteration count of the MTCBO-LR algorithm as the benchmark, set as the abscissa. It used the corresponding true factor value of all algorithms as the ordinate. Our findings indicate that the MTCBO-LR algorithm outperforms other EMT algorithms regarding convergence speed across

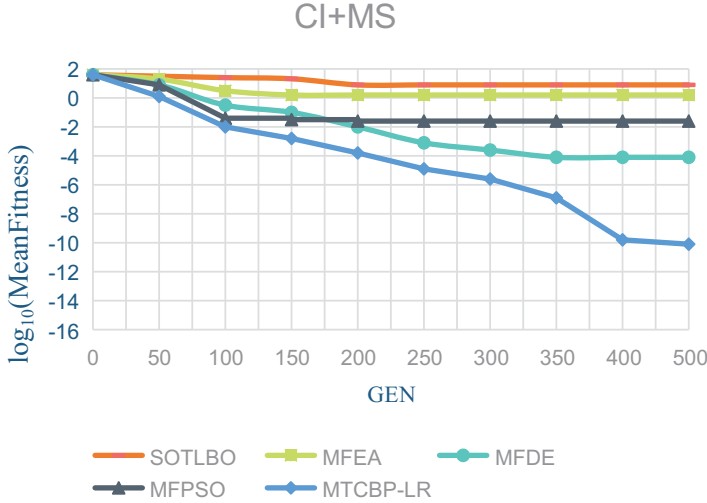

**Figure 3** **The effectiveness of five algorithms on CI+MS tasks.**

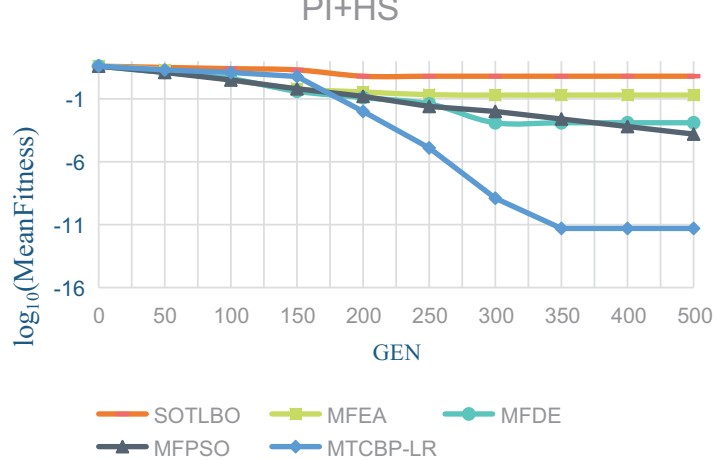

**Figure 4** **The effectiveness of five algorithms on PI+HS tasks.**

most test sets. The algorithm generates high-quality solutions within fewer than 150 iterations for the CI+HS and NI+HS test problems. This superior performance is attributed to the algorithm's multifaceted approach and enhanced accuracy and speed of convergence. In contrast, other algorithms could not attain comparable results at termination. Table 2 provides an overview of the parameter settings for the comparison algorithms.

Table 3 details the parameter configurations for both the MFEA-GHS and MFEA-AKT algorithms. To analyze the convergence performance of the MTCBO-LR algorithm, this study presents graphical visualizations of the convergence of the SOTLBO, MFEA, MFDE, MFPSO, and MTCBO-LR algorithms across five distinct sets of single-objective multitask test problems. It is worth noting that each algorithm was run for differing iterations. Using

**Table 2 Parameter settings of algorithms MFEA-GHS and MFEA-AKT.**

| MFEA-GHS | MFEA-AKT |
| --- | --- |
| Zoom ratio: sr $\in$ [0.5, 1.5] | Polynomial index of variation: 5 |
| Number of top individuals: 2 | Analog binary cross index: 2 |
| Polynomial index of variation: 5 | Arithmetic cross index: 0.25 |
| Analog binary cross index: 2 | Geometric cross index: 0.25 |

**Table 3 Performance of seven groups of single-objective multi-task benchmarking problems.**

| CI+HS | Work | MFEA-GHS | MFEA | MFEA-AKT | MTCBO-LR |
| --- | --- | --- | --- | --- | --- |
| CI+MS | Griewank | 3.85E−01 | 3.59E−01 | 1.55E−05 | 0.00E+00 |
|  | Rastrign | 1.41E−01 | 2.17E+02 | 3.99E−02 | 0.00E+00 |
| CI+LS | Ackley | 4.12E+00 | 4.63E+00 | 6.11E−05 | 2.21E−01 |
|  | Rastrign | 3.14E+02 | 2.79E+01 | 2.79E−02 | 0.01E−01 |
| PL+HS | Ackley | 2.63E+01 | 9.09E−01 | 1.23E−05 | 1.07E+01 |
|  | Schewfel | 4.73E−02 | 2.45E+02 | 3.57E−02 | 1.22E+02 |
| PI+MS | Mode | 6.17E−01 | 6.19E−01 | 2.33E−05 | 1.16E+01 |
|  | Modify | 9.02E−02 | 7.17E−02 | 5.28E−02 | 2.87E+03 |
| PI+LS | Modify | 5.83E+01 | 1.59E−01 | 3.53E−05 | 1.19E+01 |
|  | Schewfel | 1.73E+02 | 5.57E+02 | 6.10E−02 | 1.64E+03 |
| NO+HS | Mode | 8.91E−01 | 7.59E−01 | 2.25E−05 | 1.87E+02 |
|  | Ackley | 2.86E+02 | 8.17E+02 | 4.16E−02 | 2.61E−01 |

a population size of 100 and a maximum evaluation threshold of 100,000, each algorithm underwent 20 independent runs, and the convergence curve graphs for four strategies are provided.

## Global strategy effectiveness evaluation

The MTCBO-LR algorithm incorporates position. These individuals can generate new populations from their optimal historical positions. The optimal global position (gbest) selection depends on the distribution of non-dominant solutions. The unsuccessful game strategies learn from the successful ones and refer to the specific game process network. This article adopts a corresponding strategy evaluation, and convergence curves are constructed for MTCBO-LR and seven different algorithms targeting two different tasks, as shown in Figs. 5 and 6. The graphs illustrate that the convergence curve IGD value of the MTCBO-LR algorithm is higher, and the convergence speed is faster, indicating better performance.

## Discussion

In the MTTLBO algorithm, students learn from teachers and classmates in different tasks, realizing cross-task knowledge transfer. Introducing a reverse learning mechanism in the "teaching" stage can prevent convergence too fast and fall into local optimal. The

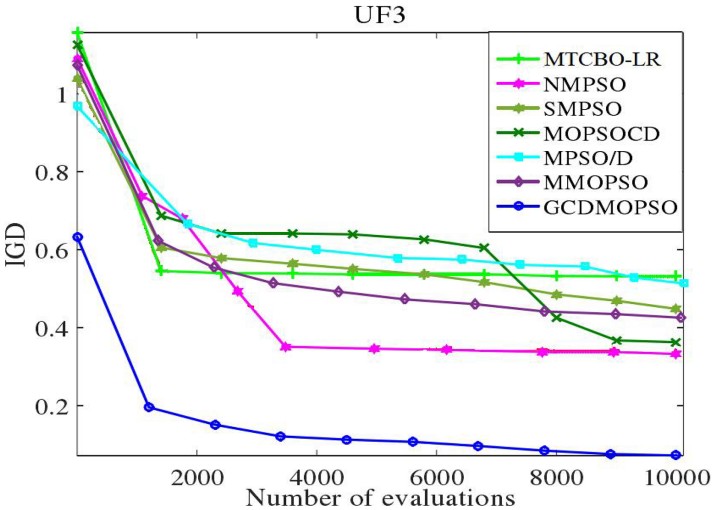

**Figure 5 Convergence curves of different algorithms on UF3.**

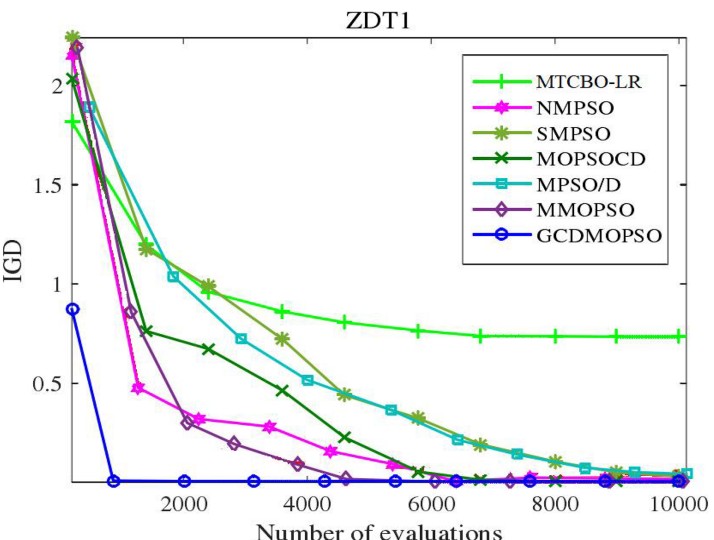

**Figure 6 Convergence curves of different algorithms on ZDT1.**

experimental results show that compared with the existing evolutionary multitasking algorithms, the MTTLBO algorithm achieves satisfactory results in single-objective multitasking optimization. In the MTTLBO-MR algorithm, each student learns from other individuals in different tasks, realizing cross-task knowledge transfer and multi-learning strategy ensures the diversity of search. In the late iteration period, the normal distribution perturbation strategy is introduced to prevent falling into local optimal effectively. Specifically, the training results of the model can better enable the members of the online collaborative learning group to help each other, ensure heterogeneity within the group, and realize the mixed grouping of learners with different cognitive levels and learning styles to realize the complementary features among the members of the learning group. The formed

online learning group contains learners with different characteristics. When completing collaborative learning tasks, various thinking modes of learners of different types interact and collide within the group to realize brainstorming, improve the efficiency of collaborative learning, and promote the cultivation of various abilities of learners.

## CONCLUSION

In this article, we propose the MTCBO multiobjective optimization strategy and the phased optimization method of MTCBO-LR, which can give different teaching strategies for students with varying learning situations to achieve better optimization results. At the same time, we also propose three different learning strategies for global effectiveness evaluation to guide the algorithm's exploration and utilization in the search process. These learning strategies can expand the exploration range, increase the exploration depth, and improve the search accuracy, thus effectively improving the optimization effect.

Our experiments showed that the MTCBO-LR algorithm showed excellent optimization results in different test problems. This demonstrates the effectiveness of our proposed multiobjective optimization strategy and phased optimization method, which can significantly improve the learning effect. The multiobjective optimization strategy of MTCBO proposed in this article and the phased optimization method of MTCBO-LR, as well as the introduction of learning strategies, provide a new idea and method for solving multiobjective optimization problems. Future research can further explore how to apply these strategies in practical problems and how to combine them with other optimization algorithms to improve the optimization effect further.

Currently, the most lacking mathematics education is reasonable and reliable teaching strategies and personalized plans. The MTCBO-LR algorithm designed in this article can effectively realize the complement process of the corresponding strategy, ensure the successful implementation and development of teaching, and achieve good teaching results.

### Funding
The authors received no funding for this work.

### Competing Interests
The authors declare that they have no competing interests.

### Author Contributions
- Jiafeng Li conceived and designed the experiments, performed the computation work, authored or reviewed drafts of the article, and approved the final draft.
- Lixia Cao performed the experiments, authored or reviewed drafts of the article, and approved the final draft.
- Guoliang Zhang analyzed the data, prepared figures and/or tables, and approved the final draft.

## Data Availability

The dataset is available in the Supplemental Files.

The data is also available at Zenodo:

Eloy y Soulaiman. (2022). Courses dataset (1.0.0) [Data set]. Zenodo. https://doi.org/10.5281/zenodo.7332123.

Code is available at Zenodo:

None. (2023). TLBO. Zenodo. https://doi.org/10.5281/zenodo.8141557.

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
