# Peer review of "Research on automatic matching of online mathematics courses and design of teaching activities based on multiobjective optimization algorithm"

_PeerJ Computer Science, doi:10.7717/peerj-cs.1501_

## Round 0.1 · original submission · Major Revisions

Please careful revise and resubmit for further consideration.

Reviewer 1 ·

Basic reporting

In order to achieve the accurate design of mathematics teaching activities, this paper proposes a design scheme MTCBO-LR based on multi-task optimization (EMTO), which can realize accurate knowledge transfer and data interaction between different teachers. Meanwhile, it combines the traditional TLBO algorithm to achieve multi-task optimization. It can provide different learning strategies for students to choose from in different stages of mathematics teaching. It simultaneously can realize self-adaptive strategy adjustment according to the actual teaching needs. However, it is not acceptable for publication in its present form. Please carefully address the issues raised in the comments.
(1) Why mathematics teaching task is a multi-objective optimization problem, and what constraints or conditions should be considered?
(2) Add more literature to prove some conclusive views, such as “Existing algorithms typically solve singular optimization tasks in isolation and seldom leverage knowledge gained from one task to help solve another”;
(3) The ranking of literature does not correspond to the name of the author, which needs to be paid attention to;
(4) The headings of Sections 3.1 and 3.2 are too long, and they are difficult to understand;
(5) Briefly state the similarities and differences between different learning strategies (Section 3.2);
(6) Throughout the methodology, there is a lack of logical connection between the various titles. For example, why is the "Multi-objective Optimization Problem" placed at the end of this chapter?
(7) The distribution of paragraphs in each chapter is still not reasonable, and the author needs to distinguish the relationship between research methods and research objectives correctly;
(8) Discussion section needs to be a coherent and cohesive set of arguments that take us beyond this study in particular and help us see the relevance of what the authors have proposed;
(9) Author needs to contextualize the findings in the literature and be explicit about the added value of the study towards that literature.

Experimental design

In order to achieve the accurate design of mathematics teaching activities, this paper proposes a design scheme MTCBO-LR based on multi-task optimization (EMTO), which can realize accurate knowledge transfer and data interaction between different teachers. Meanwhile, it combines the traditional TLBO algorithm to achieve multi-task optimization. It can provide different learning strategies for students to choose from in different stages of mathematics teaching. It simultaneously can realize self-adaptive strategy adjustment according to the actual teaching needs. However, it is not acceptable for publication in its present form. Please carefully address the issues raised in the comments.
(1) Why mathematics teaching task is a multi-objective optimization problem, and what constraints or conditions should be considered?
(2) Add more literature to prove some conclusive views, such as “Existing algorithms typically solve singular optimization tasks in isolation and seldom leverage knowledge gained from one task to help solve another”;
(3) The ranking of literature does not correspond to the name of the author, which needs to be paid attention to;
(4) The headings of Sections 3.1 and 3.2 are too long, and they are difficult to understand;
(5) Briefly state the similarities and differences between different learning strategies (Section 3.2);
(6) Throughout the methodology, there is a lack of logical connection between the various titles. For example, why is the "Multi-objective Optimization Problem" placed at the end of this chapter?
(7) The distribution of paragraphs in each chapter is still not reasonable, and the author needs to distinguish the relationship between research methods and research objectives correctly;
(8) Discussion section needs to be a coherent and cohesive set of arguments that take us beyond this study in particular and help us see the relevance of what the authors have proposed;
(9) Author needs to contextualize the findings in the literature and be explicit about the added value of the study towards that literature.

Validity of the findings

In order to achieve the accurate design of mathematics teaching activities, this paper proposes a design scheme MTCBO-LR based on multi-task optimization (EMTO), which can realize accurate knowledge transfer and data interaction between different teachers. Meanwhile, it combines the traditional TLBO algorithm to achieve multi-task optimization. It can provide different learning strategies for students to choose from in different stages of mathematics teaching. It simultaneously can realize self-adaptive strategy adjustment according to the actual teaching needs. However, it is not acceptable for publication in its present form. Please carefully address the issues raised in the comments.
(1) Why mathematics teaching task is a multi-objective optimization problem, and what constraints or conditions should be considered?
(2) Add more literature to prove some conclusive views, such as “Existing algorithms typically solve singular optimization tasks in isolation and seldom leverage knowledge gained from one task to help solve another”;
(3) The ranking of literature does not correspond to the name of the author, which needs to be paid attention to;
(4) The headings of Sections 3.1 and 3.2 are too long, and they are difficult to understand;
(5) Briefly state the similarities and differences between different learning strategies (Section 3.2);
(6) Throughout the methodology, there is a lack of logical connection between the various titles. For example, why is the "Multi-objective Optimization Problem" placed at the end of this chapter?
(7) The distribution of paragraphs in each chapter is still not reasonable, and the author needs to distinguish the relationship between research methods and research objectives correctly;
(8) Discussion section needs to be a coherent and cohesive set of arguments that take us beyond this study in particular and help us see the relevance of what the authors have proposed;
(9) Author needs to contextualize the findings in the literature and be explicit about the added value of the study towards that literature.

Additional comments

In order to achieve the accurate design of mathematics teaching activities, this paper proposes a design scheme MTCBO-LR based on multi-task optimization (EMTO), which can realize accurate knowledge transfer and data interaction between different teachers. Meanwhile, it combines the traditional TLBO algorithm to achieve multi-task optimization. It can provide different learning strategies for students to choose from in different stages of mathematics teaching. It simultaneously can realize self-adaptive strategy adjustment according to the actual teaching needs. However, it is not acceptable for publication in its present form. Please carefully address the issues raised in the comments.
(1) Why mathematics teaching task is a multi-objective optimization problem, and what constraints or conditions should be considered?
(2) Add more literature to prove some conclusive views, such as “Existing algorithms typically solve singular optimization tasks in isolation and seldom leverage knowledge gained from one task to help solve another”;
(3) The ranking of literature does not correspond to the name of the author, which needs to be paid attention to;
(4) The headings of Sections 3.1 and 3.2 are too long, and they are difficult to understand;
(5) Briefly state the similarities and differences between different learning strategies (Section 3.2);
(6) Throughout the methodology, there is a lack of logical connection between the various titles. For example, why is the "Multi-objective Optimization Problem" placed at the end of this chapter?
(7) The distribution of paragraphs in each chapter is still not reasonable, and the author needs to distinguish the relationship between research methods and research objectives correctly;
(8) Discussion section needs to be a coherent and cohesive set of arguments that take us beyond this study in particular and help us see the relevance of what the authors have proposed;
(9) Author needs to contextualize the findings in the literature and be explicit about the added value of the study towards that literature.

Reviewer 2 ·

Basic reporting

In this paper, the authors put forward the multi-objective optimization strategy and multi-objective strategy phased optimization method, which can give different teaching strategies according to different situations of students with different teaching strategies. The proposed three different learning strategies can expand the exploration scope, increase the exploration depth, improve the search accuracy, so as to effectively improve the optimization effect. It provides a new idea and method for solving multi-objective optimization problem.

The author lists a large number of documents in the literature review, but their numbers do not correspond to the information at the end of the manuscript, which needs to be checked thoroughly;
1. Clarify the author's work in the abstract, do not use some general description, such as "Multi-objective optimization has emerged as a crucial solution in various fields."
2. More quotes need to be added to the introduction.
3. Some sentences are incomprehensible, “Enable students to escape from local optima and utilizes a rank-based selection strategy?”

Experimental design

Authors should pay more attention to the introduction of data sets, and if necessary, should add the process or method of data preprocessing, which will help other scholars to reference and learn from this study.

Validity of the findings

Update and clearly state the major contributions of the work after the introduction to avoid overlapping with existing research conclusions;

Additional comments

1) The name of picture 1 is missing, please add.
2) Add an opening paragraph at the beginning of Section 4.2.

---

## Round 0.2 · accepted · Accept

Congratulations, your paper is acceptable after the improvement and subsequent recommendations by the experts.

Reviewer 1 ·

Basic reporting

The authors have made the raised comments.

Experimental design

The authors have made the raised comments.

Validity of the findings

The authors have made the raised comments.

Additional comments

The authors have made the raised comments.

Reviewer 2 ·

Basic reporting

The authors have accommodated the changes suggested by the reviewers

Experimental design

The authors have accommodated the changes suggested by the reviewers

Validity of the findings

The authors have accommodated the changes suggested by the reviewers

Additional comments

The authors have accommodated the changes suggested by the reviewers